# Risk of Endometriosis and Subsequent Ovary and Breast Cancers in Nurses: A Population-Based Cohort Study in Taiwan

**DOI:** 10.3390/ijerph16183469

**Published:** 2019-09-18

**Authors:** Hsing-Chi Hsu, Kai-Yu Tseng, Hsiang-Chi Wang, Fung-Chan Sung, Wei-Fen Ma

**Affiliations:** 1Department of Public Health, China Medical University College of Public Health, Taichung 404, Taiwan; chiqueens@gmail.com; 2School of Nursing, Central Taiwan University of Science and Technology, Taichung 404, Taiwan; 107179@ctust.edu.tw; 3Management Office for Health Data, China Medical University Hospital, Taichung 404, Taiwan; vwang.lk@gmail.com (H.-C.W.); fcsung1008@yahoo.com (F.-C.S.); 4College of Medicine, China Medical University, Taichung 404, Taiwan; 5Department of Health Services Administration, China Medical University College of Public Health, Taichung 404, Taiwan; 6School of Nursing, China Medical University, Taichung 404, Taiwan; 7Department of Nursing, China Medical University Hospital, Taichung 404, Taiwan

**Keywords:** endometriosis, breast cancer, nurses, ovarian cancer, retrospective cohort study

## Abstract

Background: Endometriosis has been associated with the subsequent development of ovarian and breast cancers. This study evaluated whether nurses were at increased risks of developing endometriosis and subsequent ovarian and breast cancers. Methods: From Taiwan National Health Insurance Research Database during 2000 to 2011, we established 3 study cohorts, consisting of 23,801 nurses, 11,973 other hospital employees, and 143,096 general women free of endometriosis and cancer. Women in all cohorts were followed to the end of 2011 to measure the occurrences of endometriosis and subsequent ovarian and breast cancers. The incident endometriosis cases and related hazard ratio (HR) and 95% confidence interval (CI) were calculated. The incident cases of ovarian cancer and breast cancer and related odds ratio were calculated. Results: The incidence of endometriosis was the highest in the nurse cohort (4.23 per 100, *n* = 966) followed by other health professionals (3.74 per 100, *n* = 427) and control cohort (3.06 per 100, *n* = 4193), with adjusted hazard ratios of 1.28 (95% CI = 1.20–1.38) and 1.13 (95% CI = 1.02–1.25), respectively, comparing to controls. Among those who developed endometriosis, nurses had higher subsequent ovarian cancer and lower breast cancer, but not significant. Conclusions: Nurses are at a higher risk of developing endometriosis. However, the link between endometriosis and subsequent cancers is weak.

## 1. Introduction

A systematic review showed that endometriosis is one of the most common gynecological diseases among women of childbearing age, with the prevalence rates range 2–10% among countries [1,2]. Causes of endometriosis have been associated with menstrual reflux, hormonal imbalance, heredity, specific genes, and environment and immune anomaly, but remain unclear [2,3,4]. The main clinical symptoms of endometriosis include abnormal bleeding, dysmenorrhea, chronic pelvic pain, and infertility [1,2,5]. Up to 50% of endometriosis women of childbearing age suffer from infertility [6], and these symptoms lead to negative psychological effects [7].

Women with endometriosis are at an increased risk of developing other complications. Previous studies have found that women with endometriosis are at an increased risk of epithelial ovarian cancer, with hazard ratios (HRs) of epithelial ovarian cancer range from 2.01 to 4.56 [8,9,10].

Clinical nursing is a high-stress work. Nurses need to deal with emergency cares for patients with the challenge of lifestyle change. Approximately, 85% nurses in Taiwan have reported that working stress is the major risk factor to threaten their health [11]. Marino et al. have indicated that health professionals, particularly nurses, were at increased risks of developing endometriosis, with an adjusted odds ratio (OR) of 1.60 (95% CI = 1.00–2.55) [12]. In addition, a recent study in Taiwan found that the nurses have increased risks than general female population for breast cancer (standardized incidence ratio (SIR) = 1.2) and ovarian cancer (SIR = 1.8) [13]. An American Nurse Health study also found nurses with endometriosis are at an increased risk of cancer with a relative risk of 1.81 for ovarian cancer [14]. Another study in Taiwan also found that the incidence of cancer in nurses with endometriosis increases with age [13].

Women with endometriosis are characterized with elevated estrogen and subsequently are at increased risk of breast cancer, ovarian cancer, and benign breast diseases [15,16]. Whether the presence of endometriosis aggravating the risk of ovarian and breast cancers in nurses has not been well investigated among populations. We, therefore, attempted to use the insurance claims data of Taiwan to evaluate the risk of developing endometriosis and subsequent risk of developing ovarian and breast cancers, comparing among nurses, other healthcare professionals, and non-healthcare professionals.

## 2. Materials and Methods

### 2.1. Data Sources

The present study used the National Health Insurance Research Database (NHIRD) of Taiwan from 1996 to 2011. Demographic status of insured population, treatment and medications received, and cost of care are available in the database. Diseases were coded with the International Classification of Diseases, Ninth Revision, Clinical Modification (ICD-9-CM). A subset Registry for Medical Personnel database containing the information of healthcare professionals was used to identify nurses and other types of healthcare professionals. The said information was gathered from the Registry for Contracted Medical Facilities, Registry for Prescription Files, and Medical Claims Data in the NHIRD. The personal information included identification, sex, birth date, employer, and dates of employment initiation and termination. To protect personal privacy, all identifications in the NHIRD data were encrypted by the National Health Research Institutes. The study was approved by the Research Ethics Committee, China Medical University and Hospital (IRB permits number: CMUH-104-REC2-115).

### 2.2. Study Population

From the Registry for Medical Personnel database, we identified 23,810 nurses aged 20–54 years without history of endometriosis and cancer. From 55,323 females of other healthcare professionals (physicians, dentists, pharmacists, dietitians, radiologists, and physical therapists), we randomly selected 11,973 women as “other hospital employees” cohort, frequency matched with the nurse cohort by age, without the history of endometriosis and cancer. We further randomly selected 143,096 women without history of endometriosis or cancer from general female population frequency matched with the nurse cohort and other professional cohorts by age as the control cohort.

### 2.3. Study Outcome and Comorbidity

The primary outcome of the study was endometriosis defined by ICD-9-CM (code 617), newly diagnosed in outpatients or inpatients data after the baseline date. Women in the 3 cohorts were followed up to the date of endometriosis diagnosis, withdrawal from insurance, or the end of 31 December, 2011. The study further followed women who had developed endometriosis for the occurrence of ovarian and breast cancers as secondary outcomes. Claim records of pelvic inflammatory disease (PID) (ICD-9-CM code 614), cardiovascular disease (CVD) (ICD-9-CM code 410–414, 428, 430–438, 440–448), infertility (ICD-9-CM code 606, 628), diabetes mellitus (ICD-9-CM code 250), chronic liver disease (ICD-9-CM code 275.0), leiomyoma of uterus (ICD-9-CM code 218), and autoimmune disease (ICD-9-CM code 710.0, 714.0, 340) were identified at the baseline as comorbidities.

### 2.4. Statistical Analysis

Baseline characteristics presented as categorical variables were tested with $\chi^{2}$ test. ANOVA test was used to examine means among cohorts. Incident endometriosis cases were estimated for the 3 cohorts and the Cox proportional hazards regression analysis was applied to estimate the hazard ratio (HR) of endometriosis and related 95% confidence interval (CI). The adjusted HR (aHR) was estimated using multivariable regression analysis by including study cohort, age, and comorbidities with significant crude HR (cHR). Subsequent cases of ovarian cancer and breast cancer were estimated for women who had developed endometriosis among the 3 study cohorts. We used logistic regression analysis to calculate the odds ratio (OR) of developing ovarian cancer and breast cancer among women with endometriosis by study cohort and age. We used SAS statistical analysis software, Version 9.4 (SAS Institute Inc., Cary, North Carolina) to conduct data analyses with the significance level of 0.05.

## 3. Results

The study population had a mean age of 34 years, and approximately 42% of women aged 30–39 years (Table 1). Baseline comorbidities were more prevalent in the control cohort than in other two cohorts, particularly of cardiovascular disease and diabetes mellitus.

Table 2 shows that the occurrence of endometriosis was the highest in the nurse cohort (4.06 per 100, *n* = 966), followed by the other hospital employees (3.57 per 100, *n* = 427) and the control cohort (2.93 per 100, *n* = 4193). The aHRs of endometriosis were 1.28 (95% CI, 1.20–1.38) (*p* < 0.0001) for the nurse cohort and 1.13 (95% CI, 1.02–1.25) (*p* = 0.018) for other hospital employees, compared to the control cohort. Women aged 30–39 years had the highest aHR of endometriosis among age groups, compared with the youngest group. Women with pelvic inflammatory disease and infertility were also at a higher hazard of developing endometriosis.

Table 3 shows that 10 ovarian cancer cases and 37 breast cancer cases subsequently occurred among women who had developed endometriosis during the follow-up period. The nurse cohort had a higher incidence of ovarian cancer and a lower incidence of breast cancer than other 2 cohorts, but not significant, compared with the control cohort. Other hospital employees had higher incident ovarian cancer and breast cancer than the control cohort had, but not significant. There was no significant variation in ovarian cancer cases among age groups. The estimated breast cancer risk increased with age, with an adjusted OR of 14.8 (95% CI = 1.90–98.7) for women aged 40–49 years.

## 4. Discussion

Our study results showed that among the 3 study cohorts, the nurse cohort had an incidence of endometriosis 14% higher than the other female health professionals or 38% higher than the control cohort, with an aHR of 1.28 relative to the control cohort. Other female health professionals also had the incidence of endometriosis significantly higher than the control cohort. The data indicate that health professionals are more likely to be diagnosed for endometriosis.

Previous studies have demonstrated that hyperlipidemia and hypertension [17], obesity, diabetes [8,18,19], and comorbidities of PID, infertility, CVD, chronic liver disease, and rheumatic diseases are prevalent in women with endometriosis [17,18]. We compared the baseline comorbidities among the three study cohorts and found that PID, CVD, and diabetes mellitus (DM) were the main comorbidities which were greater in the control cohort. It is interesting to note that PID and infertility were marked as only significant comorbidities associated with developing endometriosis in the present study (Table 2). The relationship between PID and endometriosis could be stronger for the nurse cohort and the other health professionals than the control cohort. The endometriosis risk is thus greater in the nurse cohort and other health professionals than in the control cohort. Nurses are more likely to change jobs than other health employees. Moreover, a large portion of nurses has a night-shift work style. Marino et al. also found that nurses or health aides are at particular higher risk of endometriosis [12]. They also found that working tenure and education did not have an association with an increased risk of endometriosis.

The age-specific data in this study showed that women aged 30–39 years were at the highest hazard to develop endometriosis than women aged 40–49 years, compared with the 20–29 years old group. The risk of endometriosis plummeted to 77% in women beyond 49 years. A German study evaluated 20,835 hospitalized women with endometriosis and found that 39.7% of cases were 30–39 years old or 78.4% of cases were 20–44 years old [19]. The percentage reduced to 7.1% in women aged 50 years and older, which may be associated with a decline in ovarian function leading to the atrophy of the endometrium and to the reduction of endometriosis symptoms [1]. They also noted that nurses with endometriosis were more likely to have the comorbidities of PID and infertility than nurses without endometriosis [20]. Our study demonstrated a similar relationship. It is well known that PID and endometriosis share similar symptom of pelvic pain. Some PID cases may actually a misdiagnosis of endometriosis made earlier in the patient’s life. However, it is important to note that the mean age from the onset of PID symptoms to the diagnosis of endometriosis was 10.4 years [21]. Early diagnosis of endometriosis may be delayed due to the confused or similar symptoms of PID. Therefore, early and accurate diagnosis is more crucial.

Previous studies have confirmed a correlation between endometriosis with ovarian cancer [8,9,10,14]. Our study failed to show significant difference among the 3 study cohorts in the relationships between endometriosis and ovarian cancer and between endometriosis and breast cancer. This is because of small number of ovarian cancer and breast cancer cases. The follow-up time is likely not long enough for women with endometriosis to develop these two cancers. However, our study did show that the breast cancer risk increased with age when all women with endometriosis in the 3 cohorts were pooled for the data analysis. Previous studies with a large general female population found women with endometriosis are at an increased risk of developing breast cancer after a long follow-up time [9,15]. Anifantaki et al. concluded in a review that the risk of breast cancer increases with age for women with endometriosis [22].

## 5. Limitations

This study is a population-based data to evaluate the risk of developing endometriosis and subsequent ovarian cancer and breast cancer among nurses, other healthcare employees, and general women. However, information on lifestyle, social activities, job tenure, and the level of stress at work in the study population was unavailable. We were unable to include these factors in the data analysis. Further study needs to include these factors with a larger cohort size and longer follow-up period.

## 6. Conclusions

This study demonstrated that nurses and other health workers are at a higher risk of developing endometriosis than are general women. PID and infertility could be also associated with the development of endometriosis. Being a nurse or other health worker should be a protective factor instead of an occupational risk. However, whether nurses at a higher risk for endometriosis because of stress needs investigation with more studies. Further study also requires a larger sample with a longer follow-up period to evaluate subsequent cancer risk among those with endometriosis.

## Figures and Tables

**Table 1 ijerph-16-03469-t001:** Demographic characteristics and comorbidities.

Variable	Study Cohorts						*p*-Value
	Controls		Nurses		Other Hospital Employee *		
	*n* = 143,096		*n* = 23,801		*n* = 11,973		
	*n*	%	*n*	%	*n*	%	
Age, Years ^†^							
20–29	48,816	34.1	8136	34.2	4068	34.0	0.35
30–39	60,564	42.3	10,094	42.4	5047	42.2	
40–49	27,588	19.3	4598	19.3	2299	19.2	
50–54	3504	2.45	584	2.45	292	2.44	
Unknown	2624	1.83	389	1.63	267	2.23	
Mean (SD)	34.0(7.74)		34.0(7.66)		34.1(7.52)		0.25
Comorbidities							
PID	981	0.69	145	0.61	52	0.43	0.003
Infertility	98	0.07	16	0.07	10	0.08	0.83
CVD	646	0.45	59	0.25	34	0.28	<0.0001
DM	584	0.41	32	0.13	20	0.17	<0.0001
Leiomyoma of uterus	782	0.55	133	0.56	49	0.41	0.13
Autoimmune	109	0.08	26	0.11	7	0.06	0.17

* Physician, pharmacist, dentist, dietitian, radiologist, physiotherapist. *p*-value: Chi-square test for categorical variable, ANOVA for means. ^†^ 3280 women with missing age. Abbreviation: SD: standard deviation; PID: Pelvic inflammatory disease, CVD: Cardiovascular disease, DM: diabetes mellitus.

**Table 2 ijerph-16-03469-t002:** Endometriosis cases by study cohort, age, and comorbidity and Cox method estimated hazard ratios of endometriosis.

Variable	Endometriosis	Crude		Adjusted ^†^	
	(*n* = 5586)	HR (95% CI)	*p*-Value	HR (95% CI)	*p*-Value
Occupation	*n*				
Controls	4193	1 (Reference)		1 (Reference)	
Nurses	966	1.28 (1.19, 1.37)	<0.0001	1.28 (1.20, 1.38)	<0.0001
Other hospital employees *	427	1.12 (1.01, 1.24)	0.027	1.13 (1.02, 1.25)	0.018
Age, years					
20–29	1332	1 (Reference)		1 (Reference)	
30–39	3186	1.81 (1.69, 1.92)	<0.0001	1.81 (1.70, 1.93)	<0.0001
40–49	1037	1.24 (1.15, 1.35)	<0.0001	1.25 (1.15, 1.35)	<0.0001
50–54	24	0.22 (0.15, 0.34)	<0.0001	0.23 (0.15, 0.34)	<0.0001
Comorbidities					
PID	61	1.58 (1.23, 2.03)	0.0004	1.41 (1.09, 1.83)	0.01
Infertility	11	2.70 (1.49, 4.87)	0.001	1.97 (1.07, 3.62)	0.03
CVD	20	0.95 (0.61, 1.48)	0.82		
DM	9	0.50 (0.26, 0.96)	0.04		
Leiomyoma of uterus	36	1.12 (0.81, 1.56)	0.49		
Autoimmune disease	6	1.40 (0.63, 3.11)	0.41		

*: Physician, pharmacist, dentist, dietitian, radiologist, physiotherapist. HR, Cox regression model estimated hazard ratio; Adjusted HR ^†^ estimated using multivariable analysis; CI, confidence interval. SD, standard deviation; PID, Pelvic inflammatory disease; CVD, Cardiovascular disease; DM, diabetes mellitus.

**Table 3 ijerph-16-03469-t003:** Cases of ovary cancer and breast cancer developed among women with endometriosis and odds ratio of developing cancer by study cohort and age.

Variables	Ovary Cancer										Breast Cancer					
	Cases		Crude			Adjusted			Cases		Crude			Adjusted		
	Yes	No	OR	95%CI	*p*	OR	95%CI	*p*	Yes	No	OR	95%CI	*p*	OR	95%CI	*p*
Study cohort																
Controls	6	4187	1	Reference		1	Reference		28	4165	1	Reference		1	Reference	
Nurses	3	963	2.18	(0.54, 8.73)	0.27	2.13	(0.53, 8.53)	0.27	4	962	0.62	(0.22, 1.77)	0.37	0.62	(0.22, 1.78)	0.38
Other hospital employees *	1	426	1.64	(0.20, 13.7)	0.65	1.60	(0.19, 13.4)	0.67	5	422	1.77	(0.68, 4.60)	0.24	1.97	(0.75, 5.13)	0.17
Age, years																
20–29	3	1329	1	Reference		1	Reference		1	1331	1	Reference		1	Reference	
30–39	4	3182	0.56	(0.12, 2.49)	0.44	0.57	(0.13, 2.56)	0.46	25	3161	10.5	(1.42, 77.8)	0.02	10.8	(1.46, 80.0)	0.02
40–49	3	1034	1.29	(0.26, 6.38)	0.76	1.29	(0.26, 6.43)	0.75	11	1026	14.3	(1.84, 99.8)	0.01	14.8	(1.90, 98.7)	0.01
50–55	0	24	0						0	24	-			-		

*: Physician, pharmacist, dentist, dietitian, radiologist, and physical therapist.

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
