# Peer review of "Risk of Endometriosis and Subsequent Ovary and Breast Cancers in Nurses: A Population-Based Cohort Study in Taiwan"

_ijerph, 2019, doi:10.3390/ijerph16183469_

Round 1

Reviewer 1 Report

This is an observational study that investigates whether nurses had greater occupational risks of developing endometriosis and subsequent ovarian and breast cancers. The authors had curated a nurse cohort, other health staff and a general cohort using Taiwan's database. The results showed an increased risk for endometriosis among nurse and other health staff. However, the risk for subsequent cancer seems similar among these cohorts.

Generally, the whole manuscript was carefully prepared in good writing and structure. The logic is easy to follow. There are a few points for the authors:

the rationale of studying nurse cohort should be clarified. The authors had hypothesized a night-shift effect for nurse cohort for their occupational risk for cancers. However, the hypothesis for a higher risk for endometriosis for nurses should be clarified in the introduction part. cohort selection problem. The authors had conducted a long term observation from 1996 to 2011, which is a strength for this study. However, the authors should aware that nurses are more likely to change their job than other health employees. Moreover, not all nurses had a night-shift work style. The authors should carefully address how they curated the nurse cohort in consideration of year of working experience, change job...etc. This should be doable since the authors had access to many databases. People are prone to change their job or their work style according to their health condition. Did the authors had The conclusion is very confusing. The authors had demonstrated that nurses and other health employees had a higher risk for endometriosis but suggested the finding is simply a knowledge effect which means: health workers, including both nurse and non-nurse, are more likely to be diagnosed to endometriosis. Being a nurse or other health worker should be a protective factor instead of an occupational risk. The confidence intervals for subsequent cancer risk are too wide to make a conclusion such as "among those developed endometriosis, nurses had higher subsequent ovarian cancer, lower breast cancer, but not significant." The authors should reconsider the appropriateness of their statement: "sample sizes of endometriosis were not large enough to observe the subsequent development of ovarian cancer or breast cancer". The authors should consider alternative study design (discontinuity), statistical method (report SIR instead of HR) or simply try to obtain a newer database for longer and larger samples. 

Author Response

Response to Comments from Reviewer 1

Re: ijerph-585148

Risk of endometriosis and subsequent ovary and breast cancers in nurses: A population-based cohort study in Taiwan.

Dear Reviewer:

Thank you very much for taking time to review our article and giving us very helpful comments and suggestions. We have tried our best to revise our article accordingly point by point. Changes in the text have been outlined in Red. We really appreciate your insightful comments and suggestions for our paper that helped to improve our article.

Point 1: The rationale of studying nurse cohort should be clarified. The authors had hypothesized a night-shift effect for nurse cohort for their occupational risk for cancers. However, the hypothesis for a higher risk for endometriosis for nurses should be clarified in the introduction part.

Reply: Thank you very much for your suggestions. We have addressed the hypothesis for a higher risk for endometriosis for nurses in the Introduction section (Please see Line 51-54, Introduction) and added a reference (Please see Line 232-233, Reference)

“Clinical nursing is at a high-stress at work. Nurses need to deal with emergency cares for patients with the challenge of lifestyle change. Approximately, 85% nurses in Taiwan have reported that working stress is the major risk factor to threaten their health [11]. Stressful jobs have been associated with the development of menstrual cramps, leading to menstrual reflux and endometriosis [4].Marino et al. have indicated that health professionals, particularly nurses, were at increased risks of developing endometriosis, with an adjusted odds ratio (OR) of 1.60 (95% CI = 1.00-2.55) [12].”

Point 2: cohort selection problem. The authors had conducted a long term observation from 1996 to 2011, which is a strength for this study. However, the authors should aware that nurses are more likely to change their job than other health employees. Moreover, not all nurses had a night-shift work style. The authors should carefully address how they curated the nurse cohort in consideration of year of working experience, change job...etc. This should be doable since the authors had access to many databases. People are prone to change their job or their work style according to their health condition. Did the authors had The conclusion is very confusing.

Reply: Thanks for reminding us of the work style of nurses. We have adopted your comment and discussed the issue (Please see Line 146-151). We also cited a reference related (Please see Line 232-233, Reference): “The relationship between PID and endometriosis could be stronger for the nurse cohort and the other health professionals than the control cohort. The endometriosis risk is thus greater in the nurse cohort and other health professionals than in the control cohort. Nurses are more likely to change jobs than other health employees. Moreover, a large portion of nurses have a night-shift work style. Marino et al. also found that nurses or health aides are at a particular higher risk of endometriosis[12]. They also found that working tenure and education did not appear an association with increased risk of endometriosis.”

Point 3: The authors had demonstrated that nurses and other health employees had a higher risk for endometriosis but suggested the finding is simply a knowledge effect which means: health workers, including both nurse and non-nurse, are more likely to be diagnosed to endometriosis. Being a nurse or other health worker should be a protective factor instead of an occupational risk.

Reply: Thanks for the inspirational suggestions. We have simply adopted the comment in the concluding section: “This study demonstrated that nurses and other health workers are at a greater risk of developing endometriosis than are general women. PID and infertility could be also associated with the development of endometriosis. Being a nurse or other health worker should be a protective factor instead of an occupational risk.” (please see lines 183-187, Conclusion).

Point 4: The confidence intervals for subsequent cancer risk are too wide to make a conclusion such as "among those developed endometriosis, nurses had higher subsequent ovarian cancer, lower breast cancer, but not significant." The authors should reconsider the appropriateness of their statement: "sample sizes of endometriosis were not large enough to observe the subsequent development of ovarian cancer or breast cancer". The authors should consider alternative study design (discontinuity), statistical method (report SIR instead of HR) or simply try to obtain a newer database for longer and larger samples.

Response 4: Thanks for the thoughtful comments. The Department we are working with is applying a new set of claims data with information of whole population. We will try to do additional related study. Analytical methods using SIR will be considered. Thanks for the suggestion. In the Abstract, we have stated “Among those who developed endometriosis, nurses had nonsignificantly higher subsequent ovarian cancer, lower breast cancer. Conclusions: Nurses are at a higher risk of developing endometriosis. However, the sample sizes of endometriosis were not large enough to observe the subsequent development of ovarian cancer or breast cancer. Additional study should be considered.” (Please see Line 33-37)

Reviewer 2 Report

Congratulations to the authors. This is a straightforward and understandable study with modest but interesting results.

Could some of the cases of PID actually have been a misdiagnosis of endometriosis made earlier in a patient's life? This point needs a little discussion.

Line 33: “… among those who developed endometriosis. . . “ is correct English.

Line 40: “. . . that endometriosis is one of the most common . . . “ is correct English

Line 48: “. . . lead to negative psychological effects.. .” is better English

Line 65: “subsequent” instead of “subsequence”

Line 87: ". . . without history of endometriosis or cancer . . . " is better English

Line 143: "It is interesting to note . . . " is better English

Line 153: " . . . plummeted to 77% . . . " is better English

Author Response

Response to Comments from Reviewer 2

Re: ijerph-585148

Risk of endometriosis and subsequent ovary and breast cancers in nurses: A population-based cohort study in Taiwan.

Dear Reviewer:

Thank you very much for taking time to review our article and giving us very helpful comments and suggestions. We have tried our best to revise our article accordingly point by point. Changes in the text have been outlined in Red. We really appreciate your insightful comments and suggestions for our paper that helped to improve our article.

Point 1: Could some of the cases of PID actually have been a misdiagnosis of endometriosis made earlier in a patient's life? This point needs a little discussion.

Reply:  

Thanks for the suggestion. We have adopted your comment in the discussion section (Please see Line 160-163, Discussion) and added an additional reference (Please see Line 257-259, References) to address the relation between PID and misdiagnosis of endometriosis: “It is well known that PID and endometriosis share similar symptom of pelvic pain. Some of the cases of PID may actually a misdiagnosis of endometriosis made earlier in a patient's life. However, it is important to note, the mean age from the onset PID symptoms to the diagnosis of endometriosis was 10.4 years [21]. Early diagnosis of endometriosis may be delayed due to the confused or similar symptoms of PID. Therefore, early and accurate diagnosis is more crucial.”

Point 2:

Line 33: “… among those who developed endometriosis. . . “ is correct English.

Reply: Thanks for the suggestion. We have revised accordingly.

Line 40: “. . . that endometriosis is one of the most common . . . “ is correct English.

Reply: Thanks for the suggestion. We have revised accordingly.

Line 48: “. . . lead to negative psychological effects.. .” is better English.

Reply: Thanks for the suggestion. We have revised accordingly.

Line 65: “subsequent” instead of “subsequence”.

Reply: Thanks for the suggestion. We have revised accordingly.

Line 87: ". . . without history of endometriosis or cancer . . . " is better English.

 Reply: Thanks for the suggestion. We have revised accordingly.

Line 143: "It is interesting to note . . . " is better English.

Reply: Thanks for the suggestion. We have revised accordingly.

Line 153: " . . . plummeted to 77% . . . " is better English.

Reply: Thanks for the suggestion. We have revised accordingly.

Response 2: Thanks for reviewer’s suggestions. We agreed and revised English writing in the text (Please see Line 33, 41, 47, 65, 87, 143, 154). We believe this is clearer for our readers. Thank you.

Round 2

Reviewer 1 Report

The authors had responded and revised their manuscript. However, the rationale of endometriosis and subsequent cancer remained confusing. The authors had hypothesized two independent mechanisms: a higher cancer risk because of night-shift effect and a higher risk for endometriosis because of stress. The link between endometriosis and "subsequent" cancer is weak.
